# Update on Urinary Tract Infection Antibiotic Resistance—A Retrospective Study in Females in Conjunction with Clinical Data

**DOI:** 10.3390/life14010106

**Published:** 2024-01-09

**Authors:** Cristian Mareș, Răzvan-Cosmin Petca, Răzvan-Ionuț Popescu, Aida Petca, Răzvan Mulțescu, Cătălin Andrei Bulai, Cosmin Victor Ene, Petrișor Aurelian Geavlete, Bogdan Florin Geavlete, Viorel Jinga

**Affiliations:** 1Department of Urology, “Carol Davila” University of Medicine and Pharmacy, 8 Eroii Sanitari Blvd., 050474 Bucharest, Romania; cristian.mares@drd.umfcd.ro (C.M.); razvan-ionut.popescu@drd.umfcd.ro (R.-I.P.); catalin.bulai@umfcd.ro (C.A.B.); cosmin.ene@umfcd.ro (C.V.E.); petrisor.geavlete@umfcd.ro (P.A.G.); bogdan.geavlete@umfcd.ro (B.F.G.); viorel.jinga@umfcd.ro (V.J.); 2Department of Urology, “Saint John” Clinical Emergency Hospital, 13 Vitan-Barzesti Str., 042122 Bucharest, Romania; razvanmultescu@yahoo.com; 3Department of Urology, “Prof. Dr. Th. Burghele” Clinical Hospital, 20 Panduri Str., 050659 Bucharest, Romania; 4Department of Obstetrics and Gynecology, “Carol Davila” University of Medicine and Pharmacy, 8 Eroii Sanitari Blvd., 050474 Bucharest, Romania; aida.petca@umfcd.ro; 5Department of Obstetrics and Gynecology, Elias University Emergency Hospital, 17 Mărăști Blvd., 050474 Bucharest, Romania; 6Medical Sciences Section, Academy of Romanian Scientists, 050085 Bucharest, Romania

**Keywords:** urinary tract infections, UTIs, antimicrobial resistance, AMR, uropathogens, *E. coli*, *Klebsiella*, *Enterococcus*

## Abstract

Urinary tract infections (UTIs) represent a frequent pathology among the female population that has become more and more difficult to treat in the past decade, considering the increase in antibiotic resistance—a serious global public health problem. A cross-sectional retrospective study was conducted for six months to report an update regarding the rates of resistance and susceptibility of uropathogens necessary for optimal treatment. A total of 5487 patients were screened, of which 524 (9.54%) were female patients who met the criteria for inclusion in the study. *Escherichia coli* was the most common pathogen, representing 290 cases (55.34%), followed by *Enterococcus* spp. 82 (15.64%). *Escherichia coli* presented the highest resistance to amoxicillin-clavulanic acid (R = 33.1%), followed by trimethoprim-sulfamethoxazole (R = 32.41%) and levofloxacin (R = 32.06%). The highest sensitivity rates were observed for fosfomycin (S = 96.55%), followed by imipenem (S = 93.1%). *Enterococcus* spp. showed the highest resistance to levofloxacin (R = 50.0%), followed by penicillin (R = 39.02%). The highest sensitivity was observed for fosfomycin (S = 90.24%), linezolid (S = 89.02%), and nitrofurantoin (S = 86.58%). The second most frequent Gram-negative uropathogen was represented by *Klebsiella* spp., which had the highest resistance to amoxicillin-clavulanic acid (R = 35.89%), followed by levofloxacin (R = 25.64) and trimethoprim-suflamethoxazole (R = 24.35%). The most frequently associated pathology was an episode of UTI in the previous year, followed by diabetes and chronic kidney disease. Antibiotic resistance is a serious problem for all clinicians who treat UTIs. An up-to-date knowledge of antibiotic resistance rates is a major necessity to stop its evolution. Overall, the highest resistance rates were observed for aminopenicillins, fluoroquinolones, and trimethoprim-sulfamethoxazole. The best susceptibility rates were observed for fosfomycin, nitrofurantoin, and carbapenems. Our report aims to guide clinicians whenever they are forced to prescribe antibiotics empirically.

## 1. Introduction

Urinary tract infection (UTI) represents bacterial presence at any level of the urinary tract in the presence of specific symptoms, being a frequently encountered pathology and a common reason for addressing a medical service, either ambulatory or emergency unit. This type of infection is widespread, affecting both women and men of all ages, but has a higher incidence of over 10% in the first category as opposed to 3% among men [1]. Only in the emergency department in the United States, over 3 million presentations annually are due to symptoms associated with an acute episode of UTI [2], while in the United Kingdom, up to 3% of all annual consultations are due to the same causes [3]. Worldwide, half of all women will get a UTI at some point in their lives, while UTIs are most common in subjects between the ages of 16 and 64, considering that half of them will repeat the episode in the first six months after the first infection [4].

Numerous risk factors, such as the infectious capacity of the infecting organism, the amount of the inoculum, and host characteristics, interact to cause a UTI. The colonization is the initial event that results in a UTI. The ascending route theorem is the most widely used one about colonization. The perineum is invaded by enteric bacteria, which then move up into the short urethra and up to the bladder; therefore, the local characteristic anatomy in women is the leading risk factor [5]. Other risk factors described are represented by facilitated ascent (urinary incontinence, catheterization, fecal incontinence, and vaginal and urethral mucosal atrophy), promoted colonization (sexual activity, spermicides, antibiotic treatment, estrogen depletion, and genetic factors/better bacterial adherence to mucosa), and reduced urine flow (atonic bladder, reduced fluid intake, high urine residue and urinary obstruction/stricture, and urinary stones) [6].

The most widespread bacteria in the etiology of UTIs are represented by *Escherichia coli*, *Klebsiella* spp., *Pseudomonas* spp., and *Proteus* spp. in the group of Gram-negative microorganisms, and *Enterococcus* spp. and *Staphylococcus* spp., representing the most common Gram-positive uropathogens [7,8,9]. Other less frequent bacteria in the development of a UTI are represented by *Morganella morganii*, *Citrobacter* spp., *Acinetobacter* spp., and *Streptococcus* spp. [10]. Almost 90% of all UTIs are represented by Gram-negative bacteria, and only 10% are Gram-positive [11]. While *Escherichia coli*, the most common uropathogen, is responsible for approximately 70% of all episodes of uncomplicated UTIs, while other more virulent germs such as *Pseudomonas* spp. and *Klebsiella* spp. are frequently involved in complicated UTIs [12]. An uncomplicated UTI is generally represented by the first infectious episode of the lower urinary tract in a woman without other associated risk factors, such as diabetes, immunocompromise, or neoplasia, while an upper urinary tract infection represents a complicated UTI, any infection in men, or urosepsis [13].

Antibiotic treatment is the cornerstone of any bacterial infection, including UTIs. However, in recent decades, the irresponsible use of antibiotics, both dedicated to human consumption and veterinary or agro-industrial [14], has led to a significant public health problem: the alarming increase in the resistance of bacteria to antibiotics [15,16]. The World Health Organization calls antibiotic resistance one of the leading causes of the threat to global security after climate change and terrorist attacks and wars [17]. Because more than half a million UTIs with antibiotic-resistant bacteria occur annually in the European Union, antimicrobial resistance (AMR) is periodically reported by the European Center for Disease Prevention and Control (ECDC) and WHO to take prompt and effective measures to stop this negative trend [18]. These results highlight the necessity of coordinated action to address antimicrobial resistance (AMR) throughout the WHO European Region, since AMR-producing microorganisms cannot be controlled within national borders.

According to the European Association of Urology (EAU), the judicious use of antimicrobials is essential in preventing the worsening of AMR, and European programs such as Antibiotic Stewardship are critical in better understanding the negative effect of overconsumption. At the same time, EAU recommends systematic evaluations of the local antibiotic resistance rates of uropathogens and the orientation of the treatment according to this in the double objective—the optimal and efficient treatment of UTIs and the prevention of the worsening of AMR [19]. Both worldwide and in Europe, the highest resistance rates of uropathogens were represented by fluoroquinolones, cephalosporins, and aminoglycosides, which are also the prescribed classes of antimicrobials in treating UTIs. Aminoglycosides and the trimethoprim-sulfamethoxazole combination have an average but increasing resistance, coming behind carbapenems, macrolides, and vancomycin which present a rapid and worrying increase in resistance [20,21].

Unfortunately, Romania is one of the biggest consumers of antibiotics at the European level, while public knowledge of the correct use of antibiotics is low. Therefore, recent data show how significant the impact of AMR is in this country and the need to implement the cautious use of these classes of drugs [22]. Local studies show a worrying increase in Gram-negative bacteria to fluoroquinolones, aminopenicillins, and cephalosporins, while sensitivity to carbapenems and aminoglycosides is relatively preserved [23,24]. The present study aims to evaluate the latest trends in antimicrobial resistance in female patients and to provide an update on the study of resistance and sensitivity of uropathogens in Romania, together with clinical data of patients who may represent a risk factor in antibiotic resistance UTIs. Clinical medicine requires a permanent update to guide the prescription of antibiotics empirically.

## 2. Materials and Methods

The present study was conducted retrospectively at “Prof. Dr. Th. Burghele” Clinical Hospital, Bucharest, Romania, for a period of 6 months between 1 September 2022 and 28 February 2023. A total of 5487 patients were screened, of which 524 (9.54%) female patients presented positive urine cultures and met the criteria for inclusion in the study. Patients over 18 years old, with urine cultures of more than 10^5^ CFU/mL, and single bacterial strains, were included in this study. The exclusion criteria of the current study were represented by male sex or patients under 18 years old, urine cultures with less than 10^5^ CFU/mL, two or more bacterial strains on urine culture, and patients with urinary catheters.

While demographic information was gathered for both hospitalized and non-hospitalized patients, such as age and sex, data on the study population’s history, behavior, and clinical characteristics could be extracted from the patient observation charts only for those hospitalized, considering the study’s retrospective nature.

After being given clear instructions, participants self-collected 5–10 mL of clean-catch, mid-stream urine (MSU) samples in a sterile urine container. Urine sample collection followed international safety recommendations in all cases [25]. Samples were taken within two hours of collection and sent to the Clinic’s Microbiology Laboratories in a special box for additional processing.

Standard European protocols for inoculation, incubation, and bacterial culture were followed. The bacterial growth of the samples above 10^5^ CFU/mL was considered significant and the main factor to include in the study. A 0.5 McFarland inoculum is created from the pure culture that is acquired after 24 h of seeding, and this inoculum is then used to inoculate the Mueller–Hinton Petri plate. This culture medium has nutrients that are ideal for the development of most bacteria and is free of antibiotic inhibitors. The Mueller–Hinton medium placed in cloth containing the test strain of discs with various antibiotics is applied to the surface using the disk-diffusimetric technique. The antibiotic achieves decreasing concentrations by circular diffusion in the surrounding environment. The following process was used to interpret the findings. The two critical diameters determined by the EUCAST [26] standard are compared to the growth inhibition zone’s diameter, stated in millimeters. The strain is indicated as sensitive (S) when the value is more than or equal to the upper critical diameter, and as resistant (R) when the value is less than the lower critical diameter. The identification of susceptibility and resistance to antibiotics was determined using the European Committee on Antimicrobial Susceptibility Testing (EUCAST) standards, the results being obtained by the Microbiology Department of the hospital, which were then interpreted by clinicians. Previous research has discussed bacterial culture, uropathogen identification, and antibiotic susceptibility testing that have been implemented [21,23,27,28].

## 3. Results

A cross-sectional retrospective study was conducted in one of the largest national urology centers, in Bucharest, Romania, at “Prof. Dr. Theodor Burghele” Clinical Hospital between 1 September 2022 and 28 February 2023. A total number of 5487 patients were screened, of which 524 (9.54%) were female patients who met the criteria for inclusion in the study. Most urine cultures were represented by uropathogens belonging to the Gram-negative group (417, 79.58%), followed at a distance by Gram-positive germs (107, 20.41% of the cultures). The most common bacteria in the studied group were from the Gram-negative group, represented by *Escherichia coli* (290, 55.34%), followed by *Enterococcus* spp. (82, 15.64%)—the most common Gram-positive uropathogen. In order of frequency, other bacteria found in the studied population were *Klebsiella* spp. (78, 14.88%), *Proteus* spp. (31, 5.91%), and *Staphylococcus* spp. (25, 4.77%). The least common pathogen in the studied group was from the Gram-negative group, represented by *Pseudomonas* spp.—18 (3.43%) cases.

An important characteristic of UTIs in the female population is represented by the variability of their incidence in close connection with age and the degree of sexual activity they have. In the studied group, an increase in the incidence of UTIs is observed in direct proportion to age—18–29 years: 28 (5.34%) cases; 30–45 years: 57 (10.87%) cases; 46–59 years: 117 (22.32%) cases; and over 60 years old: 322 (61.45) cases. The same linear increase is also observed separately in Gram-negative groups, as follows: 18–29 years: 19 (3.62%) cases; 30–45 years: 48 (9.16%); 46–59 years: 92 (17.55%); and over 60 years: 258 (49.23) cases; and Gram-positive, as follows: 18–29 years and 30–45 years: 9 (1.71%) cases; 46–59 years: 25 (4.77%) cases; and over 60 years: 64 (12.21%) cases. A complete picture of the division by groups, uropathogens, and age groups is represented in Table 1.

Among Gram-negative bacteria, *Escherichia coli* had the highest prevalence. Regarding its resistance patterns, *Escherichia coli* presented the highest rates to amoxicillin-clavulanic ac. (R = 33.1%), followed by trimethoprim-sulfamethoxazole (R = 32.41%) and levofloxacin (R = 32.06%). It showed medium resistance to aminoglycosides (amikacin, R = 18.62%) and cephalosporins (ceftazidime, R = 13.44%). The highest sensitivity rates were observed for this pathogen to fosfomycin (S = 96.55%), followed by carbapenems (imipenem, S = 93.1%). The second most frequent Gram-negative uropathogen was represented by *Klebsiella* spp. with the highest resistance to amoxicillin-clavulanic ac. (R = 35.89%), followed by levofloxacin (R = 25.64) and trimethoprim-suflamethoxazole (R = 24.35%). The highest sensitivity rates were highlighted for aminoglycosides (amikacin, S = 91.02%) and carbapenems (imipenem (S = 88.46%) and meropenem (S = 85.89%)).

*Proteus* spp., the most important urease-producing uropathogen, frequently associated with urinary lithiasis, the third most frequent among Gram-negative pathogens, showed the highest resistance to the trimethoprim-sulfamethoxazole combination (R = 54.83%), followed by levofloxacin (R = 48.38%). The best-maintained sensitivity among all tested antibiotics was for aminoglycosides (amikacin), where all tested strains were sensitive, followed by cephalosporins (ceftazidime, S = 93.54%) and carbapenems (imipenem, S = 87.09%). The least common pathogen among all strains evaluated was *Pseudomonas* spp., but often associated with nosocomial infections and with increased resistance to antibiotics, it presented the highest rates of resistance to fluoroquinolones (levofloxacin, R = 38.88%) and cephalosporins (ceftazidime, R = 38.88%), followed by aminoglycosides (amikacin, R = 27.77%). The best sensitivity rates were observed for carbapenems (meropenem (S = 77.77%) and imipenem (S = 66.66%)). The detailed representation of the sensitivity and resistance of Gram-negative uropathogens is presented in Table 2. The visual representation of *Escherichia coli* sensitivity and resistance is represented in Figure 1. All cases in which certain bacterial strains were not tested for the respective antibiotics were marked with “not available”.

*Enterococcus* spp., the second most frequent uropathogen among all urine cultures studied and the first in frequency among Gram-positive pathogens, showed the highest resistance to levofloxacin (R = 50.0%), followed by penicillin (R = 39.02%). Average and similar resistance rates were observed for ampicillin (R = 13.41%) and vancomycin (R = 3.41%). The highest sensitivity was observed for fosfomycin (S = 90.24%), linezolid (S = 89.02%), and nitrofurantoin (S = 86.58%). The least common Gram-positive germ, *Staphylococcus spp.* showed the highest resistance rates to penicillin (R = 56.0%), followed by trimethoprim-sulfamethoxazole (R = 28.0%) and amikacin (R = 24.0%). The highest sensitivity was observed in this case for linezolid (S = 84.0%). The detailed representation of the sensitivity and resistance of Gram-positive uropathogens is presented in Table 3.

The visual representation of *Enterococcus* spp. sensitivity and resistance is represented in Figure 2.

Global resistance rates of Gram-negative uropathogens were the highest for the combination of amoxicillin-clavulanic acid, representing R = 32.83%, followed by trimethoprim-sulfamethoxazole, representing R = 32.58%, and fluoroquinolones (levofloxacin) with R = 32.37%. The highest sensitivity rates were observed in the studied group for fosfomycin (S = 96.55%), followed by carbapenems (imipenem, S = 90.64%), cephalosporins (ceftazidime, S = 84.41%) and aminoglycosides (amikacin, S = 82.25%). In the group of Gram-positive pathogens, the highest rates of resistance were observed for penicillin and fluoroquinolones (levofloxacin), both having the highest and similar resistance (R = 42.99%), followed by trimethoprim-sulfamethoxazole (R = 28%) and aminoglycosides (amikacin, R =24.0%). Gram-positive uropathogens with the highest sensitivity were observed for fosfomycin (S = 90.24%), followed by linezolid (S = 87.85%) and ampicillin (S = 85.36%). Overall, within all the strains studied, the best sensitivity patterns were observed for fosfomycin (S = 95.16%), followed by carbapenems (imipenem, S = 90.64%) and linezolid (S = 87.85%). The detailed analysis of sensitivities and resistances by group and globally in all studied strains is represented in Table 4.

Considering the clinical findings in patients with UTIs, in the studied group, the highest incidence was represented by an episode of UTI in the last year, representing 187 (35.68%) cases—the most important clinical aspect among all the studied patients. It followed the diabetic patients, representing 148 (28.24%) cases, and chronic kidney disease with 95 (18.12%) cases, being defined as a decrease in glomerular filtration rate < 60 mL/min. Other clinically important factors present among the studied female patients were genitourinary neoplasm with 74 (14.12%) cases, urological surgical history in the last year with 68 (12.97) cases, or genital prolapse with 26 (4.96%) cases. The lowest frequencies of associated pathologies in the studied group were observed for immunocompromised patients (being defined as active neoplasia, human immunodeficiency virus, active chemotherapy treatment, or treatment with corticosteroids) with 25 (4.77%) cases and pregnancy with 7 (1.33%) cases. The graphic representation of frequently encountered associated pathologies in the studied group is represented in Figure 3.

## 4. Discussion

The antibiotic resistance of bacteria involved in UTIs is a serious global public health problem, with a trend of evolution marked especially in developed countries, with alarming reports indicating the urgent need for drastic measures. The most important step in raising awareness of this problem is the correct and sequential, continuous reporting of local resistance in key geographic points that will significantly assist clinicians in the respective area in the treatment of various bacterial infections. Specialists in urinary infections need recurrent, reliable updates on these resistance rates for an optimal and efficient fight against multiple types of infections, such as cystitis, prostatitis, or pyelonephritis, while fighting at the same time to limit the worsening of the resistance of uropathogens to the usual antibiotics and to prevent resorting to classes of reserve antibiotics.

### 4.1. The Incidence of Uropathogens in UTIs and Relation to Patients’ Age

The pathogens involved in UTIs are varied, but the most representative share is made up of a few common strains. *Escherichia coli* is the most common pathogen encountered in urinary infections, reaching an incidence of up to 90% in a Swedish study [29], with varying incidence rates between 55 and 75% reported in recent studies [30,31,32]. In the present study, it was followed by *Klebsiella* spp., *Proteus* spp., and *Pseudomonas* spp., in the group of Gram-negative urinary pathogens. Comparatively, a study published in 2019 in Uganda emphasizes *Escherichia coli* (41.9%) as the prime uropathogen, followed by *Staphylococcus aureus* (31.4%) and *Klebsiella* spp. (11.6%) [33]. Moreover, Vitus Silago et al. note the maximum incidence of *Escherichia coli* (38.3%), followed by *Enterococcus* spp. (6.6%), *Klebsiella* spp. (5.8%), and *S. haemolyticus* (7.8%), in a recent study in Mwanza and Dar es Salaam, Tanzania [34]. A similar survey from Iraq showed *Klebsiella* spp. as the second most frequent uropathogen (9.9%), and *Staphylococcus* spp. (11.5%) and *Enterococcus* spp. (5.9%) as the first and second most frequent Gram-positive uropathogens, different from the present study [35]. The least common Gram-negative uropathogen was *Pseudomonas* spp., accounting for 3.43% of all cases studied, less than in a 2018 multicenter study developed in 20 hospitals in Bulgaria, Greece, Israel, Turkey, Hungary, Italy, Spain, and Turkey [36], where the percentage was 9.6%, almost three times higher than in the current study. In the present study, *Staphylococcus* spp. was the second most frequent Gram-positive pathogen and the fifth in incidence among all uropathogens. This Gram-positive bacterium is the second most common cause of community-acquired UTIs [37]. Meanwhile, a study published this year in Saudi Arabia, which followed the incidence of uropathogens, showed a much lower incidence of only 1.08% [38] of this bacterium compared to the present study.

Considering the relationship between the age of patients and the rate of UTI incidence, a linear increase was observed, rising from 5.34% in the <30-year-old group to 61.45% in the >60 years old group. The same increasing trend was observed in multiple previous studies [39,40,41,42], observing infectious episodes more frequently in patients with other associated comorbidities, such as diabetes, renal failure, or urinary lithiasis [43]. However, other authors also present slightly different data regarding the incidence of age-related UTIs; in China, Yuan S et al. describe in a large cohort of patients a linear increase in incidence until the age of 30, followed by a slight decrease until the age of 40, and then an increase again until the age of 60, from where it gradually decreases [44]. Another work from Iraq by Salwa Y. showed an increase in the number of incident cases of infection until the age of 30, and then the incidence decreased in the following decade [45]. A recent 2021 study from India that tracked the frequency of UTIs among the female population of reproductive age highlighted a higher prevalence (41.2%) in this population group, correlating statistically with marital status and education level [46].

### 4.2. The Resistance Profile of Gram-Negative Uropathogens

*Escherichia coli*, the most common Gram-negative uropathogen, showed the highest rates of resistance to amoxacillin-clavulanic acid, trimethoprim-sulfamethoxazole, and levofloxacin. An extensive study published this year in the United States by Jennifer H Ku et al., which followed the resistance of *Escherichia coli* strains from outpatients, pointed out similar results with high resistance rates for aminopenicillins and trimethoprim-sulfamethoxazole. Also, it noted very low resistance to nitrofurantoin (<1%), similar to the current study (1.37%) [47]. Another large study on the female population published this year and developed in Germany that followed the bacterial resistance of *Escherichia coli* in UTIs [48] pointed out the highest resistance to trimethoprim-suflamethoxazole (18.3%), followed by mecillinam (12.4%) and cephalosporins (cefpodoxime, 9.5%). Compared to the present study, in which the resistance to fluoroquinolones is very high, the German study showed very low resistances regarding this pharmaceutical group (5.2%); however, it admitted that the most prescribed antibiotics in this population group are fosfomycin and fluoroquinolones. A 2016 publication from Iran [49] revealed resistance in *Escherichia coli* when considering different antimicrobial agents, such as amikacin (89.1%), significantly higher than the present results. However, there is an even greater difference for nitrofurantoin, as the Iranian research highlighted that R = 85.9% in this specific drug; the same paper admitted increasing resistance in carbapenems (meropenem), although the current work still found meager resistance in this group (R = 0.3–0.7% for both imipenem and meropenem).

*Klebsiella* spp., the second most common Gram-negative uropathogen, presented the highest antibiotic resistance to amoxicillin-clavulanic ac. and levofloxacin. A recent study from 2021 that tracked the resistance of this pathogen to antimicrobials showed similar results regarding the classes of antibiotics with the highest resistance. It showed the best sensitivity for colistin and carbapenems [50]. Another large study, that followed over 1500 urine cultures from China between 2011 and 2019 and was published in early 2021 [51], highlighted the increase in rates of resistance to carbapenems and aminoglycosides over time; however, it showed higher rates of resistance to cephalosporins (ceftazidime) between 21.8 and 35.6%, higher than those in the current study (19.2%). A recent study by Victoria Ballen et al., carried out in Barcelona and which followed the resistance of *Klebsiella* spp. from several sources of infection (urinary, respiratory, and blood), showed the highest rates of resistance of this pathogen among the uropathogenic strains, noting the highest resistance to fluoroquinolones (42.1%), much higher than the current study (25.5%), followed by amoxicillin-clavulanic ac. and trimethoprim-sulfamethoxazole [52].

*Proteus* spp., the uropathogen most frequently associated with urinary lithiasis, being a urease producer, showed the highest resistance to levofloxacin (48.3%) and trimethoprim-sulfamethoxazole (54.8%). Licai Mo et al. last year published a study developed in China [53] that tracked the sensitivity of this Gram-negative pathogen in patients with urinary stones, observing the highest sensitivity for amikacin (99.1%), followed by carbapenems (between 96 and 98%), much higher than in the present study (87.0%), and cefoperazone-sulbactam (95.9%). On the contrary, the lowest sensitivity was observed for sulfamethoxazole (33.8%), a result relatively similar to those present in the current work (41.9%), and cephalosporins (50.8%), much lower than in the current study (93.5%). A work published by M. Khan et al., which followed the incidence and resistance of *Proteus spp.* strains in UTIs in a tertiary center [54], pointed out the highest resistance to erythromycin (80%) and rifampicin (72%), while resistance to levofloxacin (60%) was also higher than the one presented in the present study (48.3%).

*Pseudomonas* spp. represented the least common uropathogen studied in the current work (3.43%), but with significant resistance rates to common antimicrobials. A Portuguese study published in the spring of this year [55], that followed the resistance of this pathogen to antibiotics and biofilm formation, pointed out the highest sensitivities to amikacin and tobramycin, and for ceftazidime and gentamicin, only a few strains were observed as resistant. A similar paper from 2021 from southwestern Nigeria [56] also showed similar susceptibility rates for carbapenems (89%) and aminoglycosides (65%), with current data. Shrestha et al. reported from Kathmandu, Nepal [57] that only 6.5% of the isolates were resistant to imipenem, despite *Pseudomonas* spp. displaying significant rates of resistance to ciprofloxacin (36.7%) and piperacillin (57.1%), among other antibiotics.

### 4.3. The Resistance Profile of Gram-Positive Uropathogens

*Enterococcus* spp., the second most common uropathogen (15.64%) and the most prevalent Gram-positive microorganism, showed the highest rates of resistance to levofloxacin (50.0%) and penicillin (39.02%), with relatively similar results regarding fluoroquinolones with a large study published last year from Poland (norfloxacin, R = 51.4%) [58]. A comprehensive review published this spring in the journal *Antibiotics* [59] highlights the high resistance rates of this pathogen, especially the increase in the incidence of vancomycin-resistant strains (VRE); the preserved effectiveness of ampicillin, but also of nitrofurnatoin and aminoglycosides, was proven due to the increased urinary concentration. Jonas Salm et al. presented in a paper from Germany published in 2021 [60] high rates of resistance to trimethoprim-sulfamethoxazole and ciprofloxacin, raising the issue of recurrent infections with this uropathogen. Since 2012, in a paper by Eugene Lin et al. from Texas, USA [61], the problem of overtreatment of bacteriuria with *Enterococcus* has been raised, considering the alarming increase in antibiotic resistance that would increase in the next decade, increasing the prevalence of VRE strains, and subsequently leading to a much more difficult treatment to administer in acute infection with this pathogen.

*Staphylococcus* spp. represented the second most frequent Gram-positive uropathogen, emphasizing the highest rates of resistance to penicillin and trimethoprim-sulfamethoxazole. Even more worrying results than the present ones were published this year in a large meta-analysis in Nigeria [62], which showed the highest resistances to penicillins (over 80%) and aminopenicillins and the trimethoprim-sulfamethoxazole combination (over 50% of strains). Similar results were found in another study published this spring in Iran by Maryam Rafiee et al. [63], noting over 85% of *Staphylococcus* strains resistant to penicillin, and showing susceptibility rates relatively similar to the present study for linezolid (84%) and levofloxacin (94%).

### 4.4. Antimicrobial Resistance Evolution in the Short and Middle Term

Knowing the resistance of uropathogens to common antibiotics is useful. Still, it is equally important to evaluate its evolution to obtain an overview of the problem and take measures to stop or slow down its growth. Four years ago, a similar study was carried out in the same center, following the female population’s resistance and sensitivity to antibiotics of uropathogens [27]. Comparatively, the situation is dynamic, with the precise observation of the negative evolution of resistance to most antibiotics [64]. For *Escherichia coli*, four years ago, the highest sensitivity was observed for levofloxacin (32.6%), followed by amoxicillin-clavulanic acid (23.9%), while currently the highest resistance is observed for amoxicillin-clavulanic acid (33.1%), followed by trimethoprim-sulfamethoxazole (32.41%) and levofloxacin (32.06%) (Figure 4).

Regarding *Klebsiella* spp., a favorable negative evolution was observed for amoxicillin-clavulanic ac. from 52.06% (2019) to 35.89% (2023), and similarly for levofloxacin from 34.57% (2019) to 25.64 (2023). *Pseudomonas* spp. was the third most common Gram-negative uropathogen in 2019, now being overtaken in incidence by *Proteus* spp. At that time, the highest resistance to levofloxacin was 33.3%, which is currently increasing to 38.8%. If in 2019, the resistance to ceftazidime was 16.6%, now it is 38.8%, and for amikacin it has increased in the same way: from 14.2% (2019) to 27.7% (2023). The most important increase was observed for *Proteus* spp. regarding levofloxacin resistance: from 18.3% (2019) to 48.3% (2023); therefore, as regards amoxicillin-clavulanic acid, resistance is decreasing: from 32.3% (2019) to 22.5% (2023); and likewise for ceftazidime: from 9.85% (2019) to 6.45% (2023).

For *Enterococcus* spp., the most common Gram-positive uropathogen, the most significant increase in resistance was observed for levofloxacin: from 39.7% (2019) to 50.0% (2023); followed by penicillin: from 29.8% (2019) to 39.02% (2023) (Figure 5). It is alarming that for vancomycin, which is considered a reserve antibiotic, an essential increase in resistance was observed from no resistant strains in 2019 to 13.4% in 2023; similarly, an alarming increase was observed for linezolid: from 1.16% (2019) to 8.53% (2023). However, it should also be emphasized that resistance decreased with ampicillin from 18.1% (2019) to 13.4% (2023).

*Staphylococcus* spp. showed an essential decrease in resistance to levofloxacin from 42.2% (2019) to 20.0% (2023), similar to penicillin which decreased from 66.6% (2019) to 56.0% (2023). The resistance also decreased significantly for trimethoprim-sulphamethoxazole from 40% (2019) to 28% (2023). Unfortunately, at the same time, resistance to other antibiotics increased, as follows: aminoglycosides from 2.2% (2019) to 24.0% (2023), linezolid from 4.4% (2019) to 8.0 (2023), and nitrofurantoin, which presented full sensitivity at that moment and currently has a resistance of 4%.

### 4.5. Limitations

The most important limitation of the present study might seem to be the relatively limited group of enrolled patients, but it is still significant, while the evaluation of other clinical findings of patients presenting with UTIs brings a more accurate global picture of the problem of these infections prevalence and its resistance profiles. Another significant limitation is that it analyzes a group of patients from a single center that concentrates patients from all over the southern part of our country. We believe that this fact overcomes this shortcoming of the present work. The third limitation in order of relevance is the fact that this study is a retrospective one, and there are no data available about the clinical evolution of the patients treated for these infections, so it is not possible to specify the success of their treatment, the recurrence rates of the infectious episodes, or the evolution of the resistance of the pathogens within the same individual in the case of a relapse of the infection.

## 5. Conclusions

Antibiotic resistance represents a major public health problem, and its negative evolution in the medium and long term puts the clinicians who treat UTIs to the test. The highest incidence of uropathogens is represented by *Escherichia coli*, with the highest resistance to amoxicillin-clavulanic acid with a significant increase in resistance to 33.1%, while resistance to levofloxacin is relatively stationary but high at 32.06%. Regarding *Enterococcus* spp., the second most frequent uropathogen in the studied population and the first in incidence among Gram-positive ones, it shows alarming increases with the usual antibiotics of up to 50.0% with levofloxacin and 39.02% with penicillin, but also with antibiotics of reserve such as vancomycin, with an increase of up to 13.4%.

The results showed encouraging resistance rates for fosfomycin and nitrofurantoin when treating uncomplicated urinary tract infections in women with moderate resistance to trimethoprim-sulfamethoxazole empirically; as recommended by the European Association of Urology guidelines, these are the first-line antibiotics when we are faced with uncomplicated UTIs, keeping the other classes of reserve antibiotics or opting for them following the result of the antibiogram when it is required. Cephalosporins and aminoglycosides have low resistance rates, being recommended in urinary infections of the upper tract; it is of utmost importance to limit the use of fluoroquinolones when possible, limiting the growth of an already alarmingly increasing resistance even more. At the same time, carbapenems in hospitals must be severely limited.

## Figures and Tables

**Figure 1 life-14-00106-f001:**
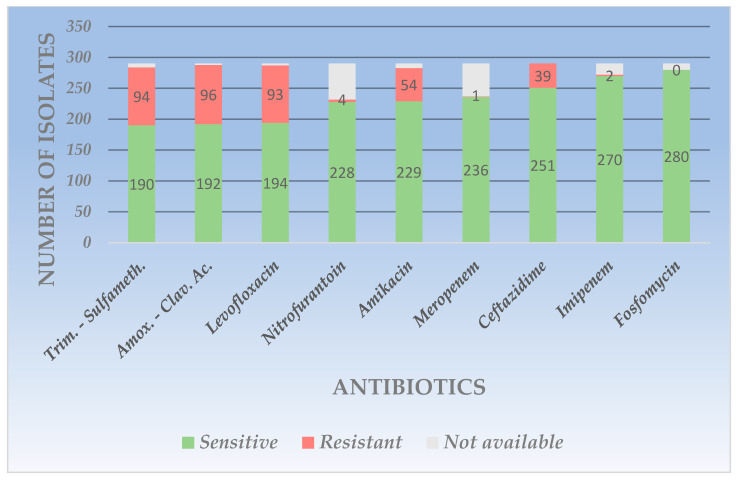
Graphic representation of resistance and sensitivity patterns for *Escherichia coli*.

**Figure 2 life-14-00106-f002:**
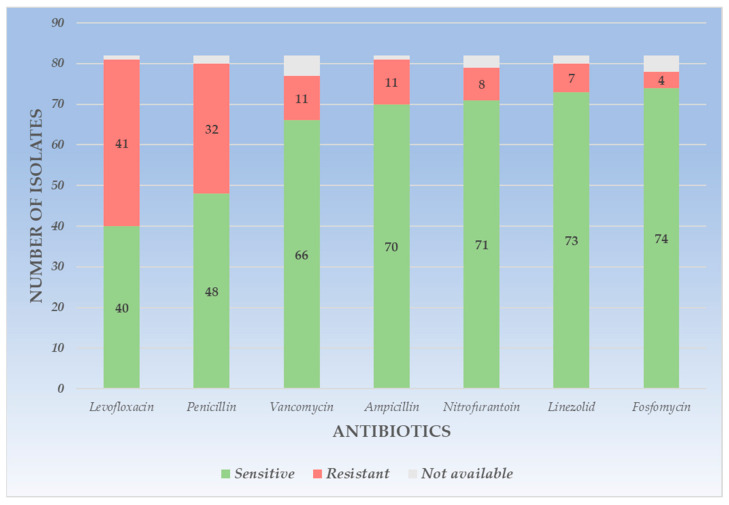
Graphic representation of resistance and sensitivity patterns for *Enterococcus* spp.

**Figure 3 life-14-00106-f003:**
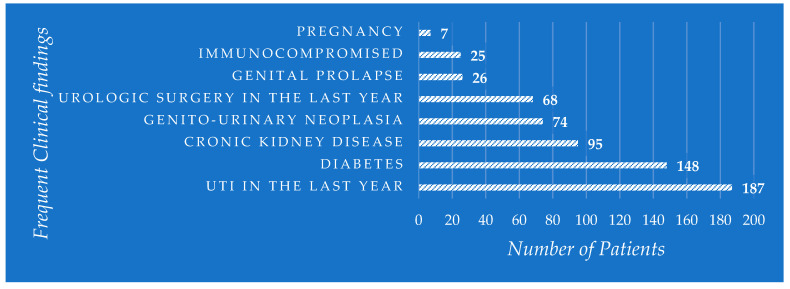
Clinical findings in the study population.

**Figure 4 life-14-00106-f004:**
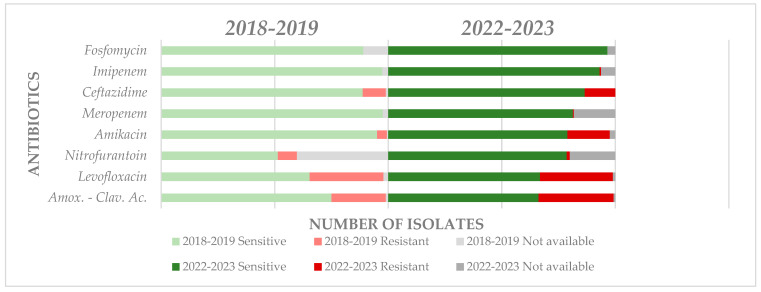
Comparative evolution of resistance and sensitivity patterns for *Escherichia coli*.

**Figure 5 life-14-00106-f005:**
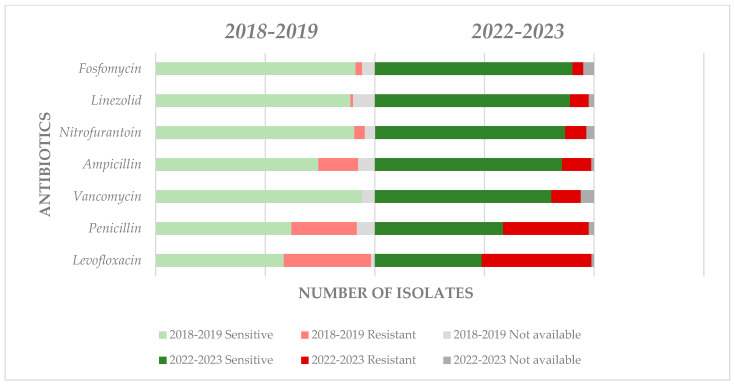
Comparative evolution of resistance and sensitivity patterns for *Enterococcus* spp.

**Table 1 life-14-00106-t001:** Uropathogens’ incidence related to age in the studied group.

	18–29 Years		30–45 Years		46–59 Years		>60 Years		Total	
	*n*	%	*n*	%	*n*	%	*n*	%	*n*	%
Gram-negative	21	3.6	48	9.1	92	17.5	258	49.2	417	79.5
*Escherichia coli*	14	2.6	31	5.9	60	11.4	185	35.	290	55.3
*Klebsiella* spp.	3	0.5	12	2.2	22	4.1	41	7.8	78	14.8
*Proteus* spp.	2	0.3	5	0.9	6	1.1	18	3.4	31	5.9
*Pseudomonas* spp.	-	-	-	-	4	0.7	14	2.6	18	3.4
Gram-positive	9	1.7	9	1.7	25	4.7	64	12.2	107	20.4
*Enterococcus* spp.	6	1.1	4	0.7	19	3.6	53	10.1	82	15.6
*Staphylococcus* spp.	3	0.5	5	0.9	6	1.1	11	2.1	25	4.7
Total	28	5.3	57	10.8	117	22.3	322	61.4	524	

*n*—number; %—percentage.

**Table 2 life-14-00106-t002:** Representation of Gram-negative uropathogens along with antibiotic sensitivity and resistance.

Antibiotics	Gram-Negative Organism Isolated																							
	*Escherichia coli*						*Klebsiella* spp.						*Proteus* spp.						*Pseudomonas* spp.					
	S		R		NA		S		R		NA		S		R		NA		S		R		NA	
	*n*	%	*n*	%			*n*	%	*n*	%			*n*	%	*n*	%			*n*	%	*n*	%		
Amikacin	229	78.9	54	18.6	7	2.4	71	91.0	7	8.9	-	-	31	100.0	-	-	-	-	12	66.6	5	27.7	1	5.5
Amoxicillin-Clavulanic ac.	192	66.2	96	33.1	2	0.6	47	60.2	28	35.8	3	3.8	22	70.9	7	22.5	2	6.4	-	-	-	-	-	-
Trimethoprim-Sulfamethoxazole	190	65.5	94	32.4	6	2.0	49	62.8	19	24.3	10	12.8	13	41.9	17	54.8	1	3.2	-	-	-	-	-	-
Ceftazidime	251	86.5	39	13.4	-	-	62	79.4	15	19.2	1	1.2	29	93.5	2	6.4	-	-	10	55.5	7	38.8	1	5.5
Fosfomycin	280	96.5	-	-	10	3.4	-	-	-	-	-	-	-	-	-	-	-	-	-	-	-	-	-	-
Imipenem	270	93.1	2	0.6	18	6.2	69	88.4	1	1.2	8	10.2	27	87.0	1	3.2	3	9.6	12	66.6	4	22.2	1	5.5
Levofloxacin	194	66.8	93	32.0	3	1.0	58	74.3	20	25.6	-	-	16	51.6	15	48.3	-	-	11	61.1	7	38.8	-	-
Meropenem	236	81.3	1	0.3	53	18.2	67	85.8	3	3.8	8	10.2	23	74.1	-	-	8	25.8	14	77.7	4	22.2	-	-
Nitrofurantoin	228	78.6	4	1.3	58	20.0	-	-	-	-	-	-	-	-	-	-	-	-	-	-	-	-	-	-

*n*—number, %—percentage; S—sensitive, R—resistant, NA—not available.

**Table 3 life-14-00106-t003:** Representation of Gram-positive uropathogens along with antibiotic sensitivity and resistance.

Antibiotics	Gram-Positive Organism Isolated											
	*Enterococcus* spp.						*Staphylococcus* spp.					
	S		R		NA		S		R		NA	
	*n*	%	*n*	%	*n*	%	*n*	%	*n*	%	*n*	%
Amikacin	-	-	-	-	-	-	16	64.0	6	24.0	3	12.0
Ampicillin	70	85.3	11	13.4	1	1.2	-	-	-	-	-	-
Trimethoprim-Sulfamethoxazole	-	-	-	-	-	-	17	68.0	7	28.0	1	4.0
Fosfomycin	74	90.2	4	4.8	4	4.8	-	-	-	-	-	-
Levofloxacin	40	48.7	41	50.0	1	1.2	19	76.0	5	20.0	1	4.0
Linezolid	73	89.0	7	8.5	2	2.4	21	84.0	2	8.0	2	8.0
Nitrofurantoin	71	86.5	8	9.7	3	3.6	16	64.0	1	4.0	8	32.0
Penicillin	48	58.5	32	39.0	2	2.4	10	40.0	14	56.0	1	4.0
Vancomycin	66	80.4	11	13.4	5	6.0	-	-	-	-	-	-

*n*—number, %—percentage; R—resistant, S—sensitive, NA—not available.

**Table 4 life-14-00106-t004:** Gram-negative and Gram-positive uropathogens’ overall resistance to common antibiotics.

Antibiotics	Gram-Negative						Gram-Positive						Total					
	S		R		NA		S		R		NA		S		R		NA	
	*n*	%	*n*	%	*n*	%	*n*	%	*n*	%	*n*	%	*n*	%	*n*	%	*n*	%
Amikacin	343	82.2	66	15.8	1	0.2	16	64.0	6	24.0	3	12.0	359	81.2	72	16.2	4	0.9
Amoxicillin-Clavulanic ac.	261	65.4	131	32.8	7	1.7	-	-	-	-	-	-	261	65.4	131	32.8	7	1.7
Ampicillin	-	-	-	-	-	-	70	85.3	11	13.4	1	1.2	70	85.3	11	13.4	1	1.2
Trimethoprim-Sulfamethoxazole	252	63.1	130	32.5	17	4.2	17	68.0	7	28.0	1	4.0	269	63.4	137	32.3	18	4.2
Ceftazidime	352	84.4	63	15.1	2	0.4	-	-	-	-	-	-	352	84.4	63	15.1	2	0.4
Fosfomycin	280	96.5	-	-	10	3.4	74	90.2	4	4.8	4	4.8	354	95.1	4	1.07	14	3.7
Imipenem	378	90.6	8	1.91	30	7.1	-	-	-	-	-	-	378	90.6	8	1.91	30	7.1
Levofloxacin	279	66.9	135	32.3	3	0.7	59	55.1	46	42.9	2	1.8	338	64.5	181	34.5	5	0.9
Linezolid	-	-	-	-	-	-	94	87.8	9	8.4	4	3.7	94	87.8	9	8.4	4	3.7
Meropenem	340	81.5	8	1.9	69	16.5	-	-	-	-	-	-	340	81.5	8	1.9	69	16.5
Nitrofurantoin	228	78.6	4	1.3	58	20.0	87	81.3	9	8.4	11	10.2	315	79.3	13	3.2	69	17.3
Penicillin	-	-	-	-	-	-	58	54.2	46	42.9	3	2.8	58	54.2	46	42.9	3	2.8
Vancomycin	-	-	-	-	-	-	66	80.4	11	13.4	5	6.0	66	80.4	11	13.4	5	6.0

*n*—number, %—percentage; S—sensitive, R—resistant, NA—not available.

## Data Availability

Data supporting the reported results are available from the authors.

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
