# Peer review of "Update on Urinary Tract Infection Antibiotic Resistance—A Retrospective Study in Females in Conjunction with Clinical Data"

_life, 2024, doi:10.3390/life14010106_

Round 1
Reviewer 1 Report
Comments and Suggestions for Authors
The submitted manuscript by Mares et al presents an update on the frequencies of various uropathogens, and their attendant antibiotic resistance patterns, in female patients suffering from confirmed urinary tract infections at a large Romanian hospital in 2023. This report is valuable in giving us a status report on antibiotic resistance rates among uropathogens at this point in time and space, which can be used both to help guide clinical practice, and as a point of comparison for different points in time, and between geographic regions. There are, however, several issues that need to be addressed to ensure scientific validity, accuracy, and relevance.
-
Full species names should be used in identifying uropathogenic isolates, unless there’s a good reason not to, or unless the clinical ID isn’t known (which seems unlikely). It isn’t appropriate to specify “Escherichia coli” (i.e. genus and species), but for all others to just indicate Klebsiella spp, Pseudomonas spp, Proteus spp, etc… There are multiple species in all of these genera, some of which are associated with UTIs and many of which aren’t, and antibiotic resistance patterns are not necessarily the same within species of a genus. In fact, they can be radically different, such as between Staphylococcus aureus, Staphylococcus saprophyticus, Staphylococcus epidermidis, and Staphylococcus haemolyticus, all of which have been associated with UTIs (with highly varying frequencies in different populations).
-
Figures 1 and 2 are redundant with information presented in Tables 2 and 3, and Figure 3 is redundant with information presented in Table 4. It isn’t necessary to present the data in both formats - the authors should choose one. I personally think the visual presentation has higher impact, but there are some issues if the authors decide to use those figures. 1) The color scheme needs to be improved - numbers are too hard to read against the background colors. 2) The label on the X axis should say “Number of isolates”, not “Number of patients” - it isn’t the patients that are resistant. 3) The figure legend should make clear what the “NA” segment actually means. 4) Some of the NA numbers are problematic - in Fig 1 for nitrofurantoin, for example, NA = 4, but the size of that colored segment seems far larger than that would warrant. It really isn’t necessary to have those numbers on the bars, but the bar size must be accurate, of course.
-
It isn’t necessary to redundantly present the same quantitative data in the text of the Results section that has also been presented in both graph and table form. We only need the data presented once. All of this makes the paper much longer than necessary. Choose one way to present the data, and be consistent with it.
-
Significant figures - there are many instances where the authors present too many significant figures on calculations. As a reminder, a calculated value shouldn’t have any more significant digits than the least of the values that goes into the calculation. Just in Table 1, all calculated percentages are shown with two digits after the decimal place, implying up to 4 significant digits, but the numerator values for those calculations may have as few as one significant digit. Please have the authors be attentive to this. Those two decimal places in the % calculation are meaningless anyway.
-
On line 162, the authors state that there is a “perfectly linear increase in the incidence of UTIs in direct proportion to age”. I challenge them to support that mathematically. It’s sufficient to say that there is “an increase in the incidence of UTIs in proportion to age”. I don’t know if the original statement was just a language issue, but there’s really no need to imply any mathematical precision.
-
In Tables 2, 3, and 4, various antibiotics appear to not have been tested against several pathogens - or at least I assume that’s what is indicated when a “-” is present. I assume this is because those antibiotics are not included in the EUCAST standards for those species (due to intrinsic resistance?), but for a non-clinician that may not be obvious. At the very least the authors should indicate in the Methods section why some antibiotics don’t show up in the data for various pathogens. I would also argue that, if all antibiotics aren’t being tested against all pathogens, then Figure 3 isn’t terribly useful. If we take Fig 3 at face value, for example, one might conclude that vancomycin is effective against > 80% of uropathogens, when in fact it doesn’t work at all against ANY of the gram-negative isolates, which are 80% of all isolates. What the figure really shows is sensitivity/resistance to the spectrum of antibiotics that have previously been deemed potentially therapeutically acceptable for that pathogen.
-
Figure 4 and the final section of the Results are not appropriate in their current form. If the authors want to examine risk factors for UTIs in their data, they need a control group of similar patients without UTIs to compare against. As logical as it might seem for many of the factors tested to increase the risk of UTI, there is no basis for making that claim unless there is a control group against which to compare. Without that, this whole section is observations and speculation, not actual epidemiological analysis of risk factors.
-
The Discussion is overly long. The entire section 4.4 can be removed. It is largely a review of risk factors for UTIs, which may be appropriate for a review article, but since I’ve already suggested that the section on risk factors in the Results be removed, it is not needed here. Section 4.5 on changes in resistance over time, on the other hand, IS interesting and important, especially since it was carried out at the same location, with presumably a similar population. I think it would be even more powerful if the authors generated a visual comparison, perhaps bar graphs with paired data showing resistance levels to the same antibiotics in 2019 and 2023.
Language usage is generally fine, although some editing is needed.
Author Response
Please find attached response.

Reviewer 2 Report
Comments and Suggestions for Authors
Life (ISSN 2075-1729)
Manuscript ID: life-2773813
Type: Article
Title: Update in Urinary Tract Infections Antibiotic Resistance – a Retrospective Study in Females in conjunction with Clinical Data
The article, titled "Update in Urinary Tract Infections Antibiotic Resistance – a Retrospective Study in Females in conjunction with Clinical Data," highlights the critical significance of bacterial antibiotic resistance in both epidemiology and clinical practice. It presents the profiles of antibiotic resistance in bacteria isolated from urinary tract infections in women. However, several limitations are acknowledged by the authors in their summary.
My comments are as follows:
Lines 48 and 60: Replace "contamination" with "presence" and "colonization," respectively.
Line 74: Correct the spelling of "Morganella morganii" by adding one "i."
Provide a detailed description of the materials and methods instead of merely citing yours four papers. It is inappropriate self-citations by authors.
Correct "Staphilococcus" to "Staphylococcus" in lines 157 and 205.
Insert a colon (":") after "46-59 years" on line 164.
In Figure 1, use "number of isolates" instead of "number of patients."
The absence of information regarding Staphylococcus species and whether they were MRS or MSS strains is a significant gap. Similarly, details about Enterococcus species and their linezolid resistance are crucial for proper result interpretation and therapeutic recommendations.
The high discrepancy in linezolid resistance among Enterococcus and Staphylococcus, as presented in the manuscript, lacks explanation.
Figure 3 essentially duplicates a portion of Table 4. Consider integrating and standardizing the antibiotic sequence throughout the article.
Discuss the prevalence of diabetes among the patient population (28.24%) and its implications. Explore if microorganism occurrence in diabetic patients aligns with the broader patient groups and if isolated strains exhibit different drug resistance patterns.
Italicize all microorganism names consistently.
Section 4.5 should elucidate the possible changes in drug-resistant bacteria prevalence, particularly the increased resistance to linezolid, vancomycin, and fluoroquinolones. Address any alterations in antibiotic prescription protocols and treatment recommendations.
Beyond describing drug-resistant microorganism prevalence, the manuscript should offer clear guidance to clinicians on which antibiotics to use to prevent further resistance escalation.

Author Response
Please find attached response.

Round 2
Reviewer 1 Report
Comments and Suggestions for Authors
My comments on the original version of manuscript are in regular type, and comments on the revised manuscript are italicized. The authors clearly did not make all of the changes that were recommended (coded in red), and the manuscript is definitely longer than necessary, but it has been improved since the first version in some important ways (coded in green).
-
Full species names should be used in identifying uropathogenic isolates, unless there’s a good reason not to, or unless the clinical ID isn’t known (which seems unlikely). It isn’t appropriate to specify “Escherichia coli” (i.e. genus and species), but for all others to just indicate Klebsiella spp, Pseudomonas spp, Proteus spp, etc… There are multiple species in all of these genera, some of which are associated with UTIs and many of which aren’t, and antibiotic resistance patterns are not necessarily the same within species of a genus. In fact, they can be radically different, such as between Staphylococcus aureus, Staphylococcus saprophyticus, Staphylococcus epidermidis, and Staphylococcus haemolyticus, all of which have been associated with UTIs (with highly varying frequencies in different populations).
-
The authors did not make these changes.
-
Figures 1 and 2 are redundant with information presented in Tables 2 and 3, and Figure 3 is redundant with information presented in Table 4. It isn’t necessary to present the data in both formats - the authors should choose one. I personally think the visual presentation has higher impact, but there are some issues if the authors decide to use those figures. 1) The color scheme needs to be improved - numbers are too hard to read against the background colors. 2) The label on the X axis should say “Number of isolates”, not “Number of patients” - it isn’t the patients that are resistant. 3) The figure legend should make clear what the “NA” segment actually means. 4) Some of the NA numbers are problematic - in Fig 1 for nitrofurantoin, for example, NA = 4, but the size of that colored segment seems far larger than that would warrant. It really isn’t necessary to have those numbers on the bars, but the bar size must be accurate, of course.
-
The authors eliminated one figure (Fig 3), but left the other redundancies.
-
The authors did change the Y axis legend for Fig 1 and 2.
-
There is no explanation of what “NA” means in these figures.
-
It isn’t necessary to redundantly present the same quantitative data in the text of the Results section that has also been presented in both graph and table form. We only need the data presented once. All of this makes the paper much longer than necessary. Choose one way to present the data, and be consistent with it.
-
The authors did not make these changes in the text - all of the redundant language is still present.
-
Significant figures - there are many instances where the authors present too many significant figures on calculations. As a reminder, a calculated value shouldn’t have any more significant digits than the least of the values that goes into the calculation. Just in Table 1, all calculated percentages are shown with two digits after the decimal place, implying up to 4 significant digits, but the numerator values for those calculations may have as few as one significant digit. Please have the authors be attentive to this. Those two decimal places in the % calculation are meaningless anyway.
-
The authors made the recommended changes.
-
On line 162, the authors state that there is a “perfectly linear increase in the incidence of UTIs in direct proportion to age”. I challenge them to support that mathematically. It’s sufficient to say that there is “an increase in the incidence of UTIs in proportion to age”. I don’t know if the original statement was just a language issue, but there’s really no need to imply any mathematical precision.
-
The authors made the recommended change.
-
In Tables 2, 3, and 4, various antibiotics appear to not have been tested against several pathogens - or at least I assume that’s what is indicated when a “-” is present. I assume this is because those antibiotics are not included in the EUCAST standards for those species (due to intrinsic resistance?), but for a non-clinician that may not be obvious. At the very least the authors should indicate in the Methods section why some antibiotics don’t show up in the data for various pathogens. I would also argue that, if all antibiotics aren’t being tested against all pathogens, then Figure 3 isn’t terribly useful. If we take Fig 3 at face value, for example, one might conclude that vancomycin is effective against > 80% of uropathogens, when in fact it doesn’t work at all against ANY of the gram-negative isolates, which are 80% of all isolates. What the figure really shows is sensitivity/resistance to the spectrum of antibiotics that have previously been deemed potentially therapeutically acceptable for that pathogen.
-
There is still no explanation of how antibiotics were selected for inclusion, but Figure 3 has been eliminated, which is an improvement.
-
Figure 4 and the final section of the Results are not appropriate in their current form. If the authors want to examine risk factors for UTIs in their data, they really need a control group of similar patients without UTIs to compare against. As logical as it might seem for many of the factors tested to increase the risk of UTI, there is no basis for making that claim unless there is a control group against which to compare. Without that, this whole section is merely observations and speculation, not actual epidemiological analysis of risk factors.
-
I still think the presentation is somewhat misleading, but at least the term “risk factors” has been removed so this is just a presentation of clinical observations.
-
The Discussion is overly long. The entire section 4.4 can be removed. It is largely a review of risk factors for UTIs, which may be appropriate for a review article, but since I’ve already suggested that the section on risk factors in the Results be removed, it is not needed here. Section 4.5 on changes in resistance over time, on the other hand, IS interesting and important, especially since it was carried out at the same location, with presumably a similar population. I think it would be even more powerful if the authors generated a visual comparison, perhaps bar graphs with paired data showing resistance levels to the same antibiotics in 2019 and 2023.
-
The authors have not reduced the length of the discussion at all - in fact they increased it, albeit with the recommended graphs comparing resistance over time.
Some copy editing is still necessary, but overall the English is acceptable.
Reviewer 2 Report
Comments and Suggestions for Authors
Dear author, I appreciate your response and explanations regarding the issues I raised in my review.
